# Recent Progress in Type I Aggregation-Induced Emission Photosensitizers for Photodynamic Therapy

**DOI:** 10.3390/molecules28010332

**Published:** 2022-12-31

**Authors:** Yuewen Yu, Hanyu Jia, Yubo Liu, Le Zhang, Guangxue Feng, Ben Zhong Tang

**Affiliations:** 1State Key Laboratory of Luminescent Materials and Devices, Guangdong Provincial Key Laboratory of Luminescence from Molecular Aggregates, School of Materials Science and Engineering, AIE Institute, South China University of Technology, Guangzhou 510640, China; 2Center for Aggregation-Induced Emission, AIE Institute, South China University of Technology, Guangzhou 510640, China; 3Shenzhen Institute of Aggregate Science and Technology, School of Science and Engineering, The Chinese University of Hong Kong, 2001 Longxiang Boulevard, Longgang District, Shenzhen 518172, China

**Keywords:** photodynamic therapy, aggregation-induced emission, intersystem crossing, type I photosensitizers, antitumor, antibacterial

## Abstract

In modern medicine, precision diagnosis and treatment using optical materials, such as fluorescence/photoacoustic imaging-guided photodynamic therapy (PDT), are becoming increasingly popular. Photosensitizers (PSs) are the most important component of PDT. Different from conventional PSs with planar molecular structures, which are susceptible to quenching effects caused by aggregation, the distinct advantages of AIE fluorogens open up new avenues for the development of image-guided PDT with improved treatment accuracy and efficacy in practical applications. It is critical that as much of the energy absorbed by optical materials is dissipated into the pathways required to maximize biomedical applications as possible. Intersystem crossing (ISC) represents a key step during the energy conversion process that determines many fundamental optical properties, such as increasing the efficiency of reactive oxygen species (ROS) production from PSs, thus enhancing PDT efficacy. Although some review articles have summarized the accomplishments of various optical materials in imaging and therapeutics, few of them have focused on how to improve the phototherapeutic applications, especially PDT, by adjusting the ISC process of organic optics materials. In this review, we emphasize the latest advances in the reasonable design of AIE-active PSs with type I photochemical mechanism for anticancer or antibacterial applications based on ISC modulation, as well as discuss the future prospects and challenges of them. In order to maximize the anticancer or antibacterial effects of type I AIE PSs, it is the aim of this review to offer advice for their design with the best energy conversion.

## 1. Introduction

Light has been used to treat diseases for thousands of years, as is well known [1]. In ancient Egypt and India, photochemotherapy with sunlight-activated psoralen was used to recolor vitiligo [2]. In 1903, Von Tappeiner discovered that topical eosin can kill cells when exposed to light and named this technique “photodynamic therapy (PDT)”. Following that, the fundamental principles of PDT were described for the first time [3,4]. In previous decades, PDT became an exceptional treatment strategy for treating patients, garnering enormous attention and rapid development due to its great spatiotemporal precision, anti-multidrug resistance qualities, and noninvasiveness when contrasted to other disease treatment modalities [5,6,7,8].

PDT is composed of three fundamental elements, a photosensitizer (PS), light, and oxygen (O_2_), to generate a therapeutic effect [9,10,11,12]. Under light radiation, an engaged PS could convert the energy of its excited state into O_2_ and surrounding substrates, resulting in toxic reactive oxygen species (ROS), thereby further causing the death of cancer cells, inducing tumor vascular arrest, and bringing about antitumor immunity, etc. The mechanism of the photodynamic effect is inherently complex; however, it is usually classified into two types, type I PDT and type II PDT, depending on different photochemical reaction processes (Figure 1). To achieve efficient PDT, PS can be excited from the ground singlet state (S_0_) to the excited singlet state (S_1_) under suitable wavelength light irradiation and then transformed to the more long-lived excited triplet state (T_1_) via intersystem crossing (ISC). For type I PDT, PSs at T_1_ react directly with the intracellular biological substrates or oxygen through the hydrogen transfer or electron transfer process and generate free radicals (type I ROS), including superoxide anions (O_2_^•−^), as well as hydroxyl radicals (OH^•−^), etc. [13]. As for type II PDT, a T_1_-state photosensitizer sensitizes the ground-state O_2_ to highly cytotoxic singlet oxygen (^1^O_2_) via energy transfer [14]. More importantly, type I PDT is more therapeutically effective in the hypoxic microenvironment than type II PDT because oxygen can be regenerated by intracellular superoxide dismutase (SOD)-mediated disproportionation [15,16]. In this regard, great efforts have been made to develop type I PSs for PDT.

PS plays a key role in the three important components of PDT [17,18]. Since the hematoporphyrin derivative (HpD) was reported as the first PS in the 1960s, numbers of porphyrin-based PSs and new PSs (e.g., acridine orange, mTHPC, and 5-ALA) have been created and granted approval by the Food and Drug Administration (FDA) for clinical practice [19,20,21,22]. Unfortunately, many of these reported PSs experience the aggregation-caused quenching (ACQ) effect because of their planer architectures and significant overall stacking interaction, which reduces both their fluorescence emission and ROS generation, thus further largely undermining the PDT efficiency [23,24,25]. Therefore, it is extremely desirable to develop novel PSs with uncompromised or even enhanced photosensitization capability in aggregate. Encouragingly, the discovery of aggregation-induced emission fluorogens (AIEgens) opens up fresh possibilities for PDT and fluorescence. The concept of AIE was first proposed by Tang in 2001 [26]. Diametrically opposite to the conventional ACQ fluorophores, AIEgens show minimal emission in good solvent but emit strong fluorescence in aggregate due to the largely suppressed heat dissipation pathway via restriction of intramolecular motion (RIM) [27,28,29,30,31]. In addition, some AIEgens also exhibited the aggregation-induced enhancement of ISC [32,33], further enhancing their capacity of ROS generation and thus making them the ideal candidates of PSs in PDT [34,35,36].

Up to now, there has been a partial review of papers, summarizing the outstanding achievements of AIEgens in imaging and therapy [37,38,39,40,41]; however, hardly any of them have concentrated on ways to modify the energy transfer or electron transfer processes of organic molecules connected to ISC in order to enhance optical characteristics and biological applications. Therefore, based on the above discussion and reasons, it is time to thoroughly summarize the progress of AIEgens for type I PDT. In this review, we highlight the most recent developments in type I AIE-active PSs for biomedical applications. Initially, we will give a summary of the recent molecular strategies for improving efficient type I PDT property, including donor–acceptor (D-A) effect, cationization engineering strategy, and polymerization. In addition, the PDT uses of type I AIE PSs are also examined in detail in terms of both photodynamic antitumor and antimicrobial applications as the main component of this review, including single PDT, PDT-photothermal therapy (PTT) synergistic treatment, PDT-chemodynamic therapy (CDT) synergistic treatment, PDT-gas therapy synergistic treatment and PDT-immunotherapy. Finally, a brief overview and future prospects are then covered. The purpose of this review is to provide recommendations for the thoughtful design of type I AIE PSs with required energy transfer or electron transfer to achieve the maximum effectiveness of PDT-mediated antitumor, antimicrobial as well as other biomedical applications. We believe this review offers a timely insight into the relationship between the molecular architectures, photophysical mechanisms, and features of organic AIE-active PSs and biomedical applications, which shall stimulate more exciting development in type I PDT.

**Figure 1 molecules-28-00332-f001:**
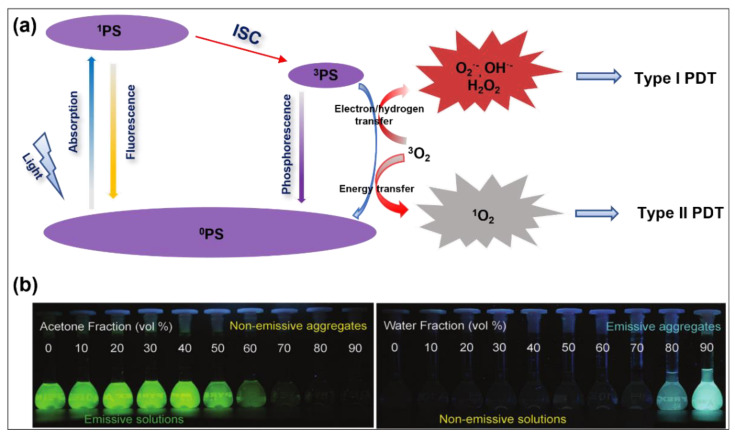
(**a**) Schematic of a Jablonski diagram showing ROS generation processes for organic PSs. (**b**) Comparison of fluorescence changes of ACQ (fluorescein isothiocyanate, **FITC**) and AIE (tetraphenylethylene, **TPE**) molecules in acetone (good solvent) and water (poor solvent) mixtures with different water fractions. Reprinted with permission from [39].

## 2. Molecular Design Strategies for Facilitating Type I PDT Processes

As mentioned above, PS is the most critical part of PDT as its performance directly determines the final treatment outcome. Although several porphyrin/ phthalocyanine/chlorin-based PSs have been approved by the FDA for clinical practice, scientific researchers and clinicians have not stopped researching and are committed to exploring new and more efficient PSs to enhance the therapeutic effects. Modulation of ISC was found to be an effective fundamental strategy to develop PSs. Generally, ISC is a spin-forbidden, nonradiative energy transition process, and the following equation can be used to describe its rate constant (*k_ISC_*) [42]:$$k_{ISC}\propto \frac{{\langle T1\left|H_{SO}\right|S1\rangle }^{2}}{{\left(\Delta {Ε}_{\mathrm{ST}}\right)}^{2}}$$
where Δ*E*_ST_ and *Hso* represent the singlet-triplet energy gap and the Hamiltonian for the spin orbital coupling (SOC), respectively. This equation indicates that for an efficient ISC, a small Δ*E*_ST_ and/or large SOC are advantageous. Therefore, it is important to facilitate the ISC processes through rational molecular engineering strategies to enhance the triplet state formation and hence ROS generation. Additional efforts are also needed to improve the electron separation and transfer ability for these PSs to generate highly toxic radicals, namely type I ROS. Up to now, some exciting research work on facilitating the ISC and electron transfer processes has emerged, which will be discussed in the following sections. 

### 2.1. Donor–Acceptor Effect

Despite the fact that heavy atoms can improve SOC and facilitate the ISC process [43,44], their practical applications in biological systems are limited due to heavy atom toxicity. The donor (D)–acceptor (A) structure is commonly employed to adjust the electronic bandgaps of organic fluorophores [45]. The D-A system allows for a larger separation in the distribution of the highest occupied molecular orbital (HOMO) as well as the lowest unoccupied molecular orbital (LUMO), resulting in a much smaller Δ*E*_ST_ and more efficient ISC [46,47]. As a result, an overwhelming amount of AIE PSs with type I photoreaction mechanism have been designed and developed based on the D-A molecular design strategy. The pioneered work of type I AIE PSs was presented by Zhao et al. in 2020. They developed two type I AIE PSs with a D-A structure based on a phosphindole oxide (PIO) core as an electron-accepting moiety (Figure 2) [48]. The iconic AIE moiety triphenylamine (TPA) and electron-withdrawing pyridine (Py) units were conjugated to the PIO core at different positions to afford ***α*-TPA-PIO** as well as ***ꞵ*-TPA-PIO**, respectively. Moreover, importantly, the strong electron affinity capability of PIO core aids in the attraction and stabilization of external electrons. Thus, when excited by light, the phosphine center attracts external electrons and breaks the phosphine–oxygen double bond to generate a radical anion, which then transfers electrons to the surrounding substrate to form OH^•−^ radicals. Hence, both ***α*-TPA-PIO** and ***ꞵ*-TPA-PIO** exhibited excellent type I ROS, especially hydroxyl radical generation. It is worth noting that ***ꞵ*-TPA-PIO** exhibits a significantly greater type I ROS-generating ability and ideal antihypoxia activity than ***α*-TPA-PIO** both in aqueous condition and in vitro as ***ꞵ*-TPA-PIO** possesses a stronger charge transfer effect and electron-accepting character and, hence, a more efficient ISC process than ***α*-TPA-PIO**.

In 2020, Wang et al. further comprehensively studied how electron donors affect the type I ROS generation via tuning the ISC and intramolecular charge transfer (ICT) processes. In this work, they designed and synthesized four NIR anion-π^+^ AIE PSs based on the D-A effect (Figure 2) [49]. TPA/methoxy-substituted TPA (MTPA) and benzo-2,1,3-thiadiozole (BZ)/naphtho [2,3-c] [1,2,5]thiadiazole (NZ) were employed as D and auxiliary D, respectively, and further coupled with styrylpyridine cation (as A). Both the electron-rich heteroatoms (S, N) in BZ and NZ as well as iodide anion facilitate the supply of electrons to the excited PS, prompting the type I ROS generation capability. Additionally, as the coordinated D moiety changes from TBZ to MTBZ, TNZ, and MTNZ, the ICT effect of the corresponding molecule gradually increases, and the Δ*E*_ST_ decreases from 0.39 to 0.20, 0.16, and 0.08 eV, for **TBZPy**, **MTBZPy**, **TNZPy**, and **MTNZPy**, respectively. The largely reduced Δ*E*_ST_ thus accelerates the ISC process and improves the generation of type I ROS. The experimental results on the solvation effect demonstrated that all four molecules have a strong ICT effect. The slope values of the Stokes shift versus the solvent polarization rate for **TNZPy** (22321 cm^−1^) were also significantly larger than those for **TBZPy** (11060 cm^−1^) and **MTBZPy** (14633 cm^−1^), demonstrating that the ICT degree gradually increased with the enhanced electron donating capacity. This amplified ICT effect can further facilitate the formation of a type I photoreaction mechanism. Upon exposure to light, the introduction of rich-electron promotes the inter/extramolecular electron transfer process, which also aids in the formation of the type I photoreaction mechanism. In the last two years, Tang et al. and other groups have created a series of D-A structured type I AIE PSs for PDT [50,51,52]. One of the widely adopted strategies is to introduce synergistic donors or π-bridges such as thiophene units to elongate the conjugated backbone lengths and spatially separate the HOMO-LUMO distribution. For example, by introducing the thiophene unit in **TI**, the resultant **TSI** (thiophene as π-bridge) and **TSSI** (bi-thiophene as π-bridge) showed red-shifted absorption that is beneficial for light penetration in tissues. Moreover, the sulfur (S) atom in thiophene can be considered a heteroatom and thus can likewise promote SOC and further increase the ISC rate. As a result, **TSSI** and **TSI** nanoparticles showed 3.6-fold and 2.1-fold higher ROS generation over **TI** nanoparticles, respectively. Importantly, the hydroxyl radical production capacity of **TSSI** nanoparticles was enhanced nearly by 35-fold after light exposure. Very recently, Yang et al. developed two AIE-active type I ROS (O_2_^•−^ and OH^•−^) generators (**NS-TPA** and **NS-STPA**) based on TPA and long-chain sulfonic acid group modified naphthiazole as D and A, respectively [52]. Given that **NA-STPA** has a slightly higher SOC (0.3596 cm^−1^) than **NA-TPA** (0.3423 cm^−1^), a much lower Δ*E*_ST_ (0.37 eV) than **NA-TPA** (0.48 eV), and a longer triplet state lifetime than **NS-TPA**, it is suggested that thiophene insertion can further strengthen the ISC process and improve the ROS generation capability, particularly for type I ROS.

Very recently, Zhao et al. developed four AIE molecules with D-A-D structure based on 9,10-phenanthrenequinone and TPA as the building blocks (Figure 2) [53]. 9,10-phenanthrenequinone was employed as the strong electron-accepting group due to its active redox cycling capability. As suggested in Figure 2, the triplet PQ and nearby substrates formed the semiquinone anion radical (PQ^•−^) via photoinduced self-electron transfer, which further react with O_2_ to form Q_2_^•−^ through electron transfer. The highly toxic QH^•−^ was then obtained after a series of other cascade reactions, including SOD enzyme-mediated disproportionation reactions. In addition, PQH_2_ can be produced in the presence of some reductases (e.g., aldo-keto reductase (AKR) and NAD(P)H quinone oxidoreductase (NQO1)) via the two-electron reduction of PQ. PQH2 can interact with PQ via disproportionation reactions to obtain PQ^•−^ with the generation of H_2_O_2_. These processes together complete a redox cycle that leads to an excellent generation capacity of type I ROS. Interestingly, the special n-π^*^ transition characteristics of PQ are also beneficial for ICT and ISC processes, thus also promoting type I ROS production ability in aggregate. 

### 2.2. Cationization Engineering Strategy

Different from type II PSs that rely on energy transfer to O_2_ to generate ^1^O_2_, the generation of type I ROS requires efficient energy transfer between T_1_ PSs and surrounding substrates. In this regard, approaches that simultaneously promote the electron transfer process and provide a rich electron microenvironment appear to be effective strategies in developing type I PSs. In 2021, Feng et al. developed a cationization molecular design strategy to acquire type I AIE PSs [54]. TPA was employed as the core, and pyridine/dimethylphenylamine as well as dicyanoisophorone moiety were used as auxiliary D and A, respectively, to construct the D-A system. The cationization of dimethylaniline or pyridine converts them from D moieties to A’ part, accompanied by a structural transition of molecules from D-A (**DTPAPy**) to A-D-A’ (**DTPAPyPF_6_**) (Figure 2 and Figure 3a). The cationic PSs shown stronger AIE characteristics than the neutral precursors. As compared to **DTPAPy**, the cationic **DTPAPyPF_6_** possesses a significant segregated HOMO-LUMO distribution (Figure 3a). Furthermore, the Δ*E*_ST_ of cationic **DTPAPyPF_6_** (0.4399 eV) is lower than its neutral precursors **DTPAPy** (0.9530 eV), hinting at a much more accelerated ISC process and more efficient ROS generation ability for **DTPAPyPF_6_**. Surprisingly, the calculated natural transition orbitals (NTOs) results show a pale hole–particle separation for **DTPAPy,** suggesting the coexistence of charge-transfer (CT) and locally excited (LE) states. In contrast, cationic **DTPAPyPF_6_** possesses excellent hole–particle separation compared to **DTPAPy**, which can effectively enhance the charge separation capacity and generates electrons and holes under light irradiation. Moreover, the much higher photocurrent for **DTPAPyPF_6_** (2.3 μA/cm^2^) than **DTPAPy** (0.29 μA/cm^2^) further demonstrated cationization could enhance the charge separation capability (Figure 3b). All these features of cationization help to produce electrons and holes for further enhancing the capacity of electron transfer and generating ability of type I ROS. Very recently, the same group further verified such a cationization molecular strategy for amplifying type I ROS generation [55]. In this work, the cyano group that is readily cationized was selected as the acceptor, methoxy-substituted TPA was used as donor with the benzothiadiazole, and benzene moieties were used as the coacceptors along with π-bridge to promote HOMO-LUMO separation. The experimental results and theoretical calculation results suggested that cationization could result in stronger electron-accepting capability, easier and better electron separation and transfer, and much lower Δ*E*_ST_, thus further accelerating the ISC process and improving the product capability of type I ROS. Moreover, the introduction of methoxy groups in TPA was also found to be able to suppress type II ROS generation while promoting type I ROS as compared to the nonsubstituted ones, hinting at the important role of electron-rich methoxy group in design type I PSs.

### 2.3. Polymerization

In recent years, the use of semiconducting polymers has become increasingly widespread in biomedical application due to its outstanding advantages, such as strong light harvesting ability, good stability, and tunable properties. In 2018, both Liu and Tang et al. independently proposed the concept of polymerization-enhanced photosensitization for PDT [56,57]. Different from small molecular PSs, polymerization results in broadened energy bands for these semiconducting polymers, which therefore decrease the overall Δ*E*_ST_ by lowering the S1 band bottom and lifting the T1 band top. Moreover, the fusion or overlap between S_1_ and T_1_ bands might occur along with the increased polymerization degree, leading to multiple S_1_-T_n_ processes [58,59]. However, such polymerization has been mainly focused on developing type II PSs. Semiconducting polymers also possess the excellent charge transport capability; in this regard, polymerization shall also be applicable for designing type I PSs. Very recently, Wu et al. reported AIE-active semiconducting polymers as efficient type I PSs. They developed three distinct semiconducting polymers, including a hyperbranched polymer (**HP**), a side chain polymer (**SP**), and a main chain polymer (**MP**), by using the famed AIE-active TPA as well as anthraquinone (AQ) fractions as D and A, respectively (Figure 3b) [60]. According to their experimental findings, all of these polymers undergo both type I and type II photoreaction pathways. Interestingly, the hyperbranched polymer **HP** possesses the best type I ROS (O_2_^•−^ and OH^•−^) as well as type II ROS (^1^O_2_) generation capacity upon 530 nm laser irradiation over **MP** and **SP**. In specific, the efficiency of **HP** in generating ^1^O_2_ is 2.5- and 1.8-fold higher than that of **SP** and **MP**, respectively. The efficiency of **HP** in generating O_2_^•−^ is 9.7- and 2.1-fold higher than that of **SP** and **MP**, respectively. Importantly, only **HP** can generate OH^•−^ among these three polymers. Furthermore, the **HP** nanoparticles show 3.5-fold O_2_^•−^-, 21-fold OH^•−^-, and slightly higher ^1^O_2_- producing efficiencies over **HP** molecules, which might be attributed to aggregation-enhanced ISC. The calculated results verify that the difference between different energy levels of model of **MP**, **SP**, and **HP** after conjugated polymerization is much smaller than the Model 1 (small molecule), resulting in an increased number of ISC channels for conjugated polymers and a higher likelihood of accelerating the formation of ISC, thus promoting the production of type I ROS, especially for hyperbranched **HP**. This hyperbranched polymer strategy provides a fresh approach to the creation of type I PS for biomedical applications.

## 3. Type I AIE PSs for Antitumor Applications

PDT has been actively used as a noninvasive treatment in clinical practice for some superficial skin cancers such as skin cancer and bladder cancer. Although many PSs have been developed for tumor treatment so far, type II photosensitizers are predominant. As mentioned earlier, since type II photodynamic therapy is highly oxygen-dependent and its therapeutic effect on anaerobic tumors is inhibited, the development of low oxygen-dependent type I PDT can effectively mitigate this problem. In this section, we describe the most recent advances of AIE-active type I PSs for PDT antitumor applications.

### 3.1. Single Type I PDT for Antitumor

As elaborated above, the efficiency in PDT is expected to be enhanced because cationization can promote the type I photoreactive pathway of AIE PSs to produce more toxic type I ROS, such as OH^•−^. It is worth mentioning that the cationization also endows the mitochondria-targeting capacity of AIE PSs, which can further enhance the PDT effect due to the fact that mitochondria are the primary target of ROS during PDT. Such mitochondrial-targeted type I PDT is demonstrated with **DTPAPyPF_6_** and **DTPANPF_6_** (Figure 3c). As shown in Figure 3d,e, the cationic AIE PSs (**DTPAPyPF_6_** and **DTPANPF_6_**) have negligible toxicity to cancer cells under dark. However, the viabilities of HeLa cancer cells have a significant decrease along with increased PS concentrations after white light irradiation (20 mWcm^−2^, 10 min). In addition, these two cationic type I AIE PSs showed significant tumor inhibition in solid hypoxia tumor upon light irradiation (Figure 3f). Both in vitro and in vivo results show the excellent antitumor PDT feature of the cationic AIE PSs. Although PSs that target the cell membrane are unable to enter the nucleus, they can cause nonapoptotic cell death and indirectly affect DNA integrity, resulting in an effective anticancer effect [61,62]. In 2022, Zhao et al. reported two AIE PSs used TPA as the rotor and D moiety, and a novel electron acceptor 2-(4-methyl-8-(pyridin-4-ylethynyl) [1,3] dithiolo [4’,5’:4,5] ben-zo [1,2c] [1,2,5] thiadiazol-6-ylidene)malononitrile as a strong A moiety (Figure 2 and Figure 3g), named the resultant AIEgens with cationic **TBMPEI** and noncationic **TBMPE**, respectively [63]. Cationization could effectively reduce Δ*E*_ST_ and promote ISC while also increasing ROS generation, particularly for type I ROS. The free radical generation ability of **TBMPEI** is superior to that of some commercial PSs (Chlorin e6 (Ce6) and Rose bengal (RB)). Furthermore, the membrane-specific targeting ability of **TBMPEI** improved its potential to destroy cancer cells when exposed to light by cell necrosis, cell membrane rupture, and DNA destruction. Finally, **TBMPEI** was successfully used for fluorescence image-guided PDI in vivo with excellent therapeutic performance.

The nucleus is also critical for PDT implementation and plays a critical role in cancer cell resistance to cell death, invasion, and metastasis [64]. Wang et al. created two AIE PSs of type I (**TFMN** and **TTFMN**) (Figure 3h) for nucleus-targeted PDT [65]. **TFMN** was built with a strong D-A structure based on the TPA moiety with D and the furan moiety as an auxiliary D and π-bridge and dicyano units as A. Furthermore, TPE, a well-known AIE-active group, was introduced into **TFMN** via refined molecular structure tuning to produce **TTFMN**. The Δ*E*_ST_ value of **TTFMN** (0.20 eV) was slightly lower than that of **TFMN** (0.24 eV). In addition, it was determined that the Gibbs free energy changes of the electron transfer processes of **TFMN** and **TTFMN** were −0.218 and −0.359 eV, respectively, indicating that **TTFMN** has better ISC and ICT processes for type I ROS (OH^•−^) generation. Furthermore, PLA12k-PEG5k-TATSA, a widely used lysosomal acid-activated TAT peptide-modified amphiphilic polymer, was employed to encapsulate **TTFMN** for nucleus targeting to enhance the PDT effectiveness of type I ROS. Driven by this “good steel used in the blade” tactic, tumor growth was significantly inhibited by the precise type I PDT.

**Figure 3 molecules-28-00332-f003:**
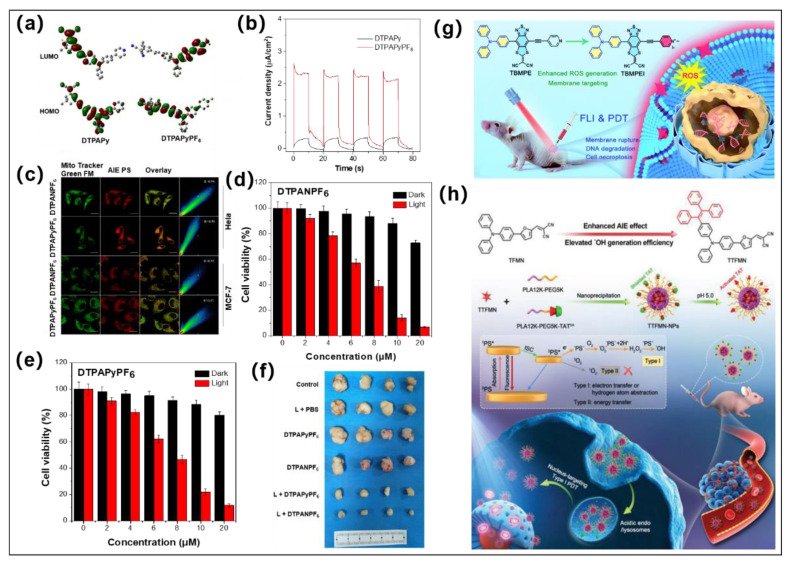
(**a**) HOMO and LUMO distribution of **DTPAPy** and **DTPAPyPF_6_**; (**b**) Photocurrent responses of **DTPAPyPF6** and **DTPAPy**. (**c**) Mitochondria-targeted confocal imaging images of **DTPAPyPF_6_** and **DTPANPF_6_**; (**d**,**e**) are viabilities of HeLa cells after treatment by **DTPANPF_6_** or **DTPAPyPF_6_** with varied concentrations under dark and light irradiation. (**f**) Tumor inhibition effect under different conditions. Reprinted with permission from [54]. (**g**) Molecular structures of **TBMPEI** and schematic diagram of **TBMPEI** for fluorescence imaging (FLI)-guided PDT antitumor. Reprinted with permission from [63]. (**h**) Molecular design of **TTFMN** and schematic diagram of **TTFMN** nanoparticles for FLI-guided PDT antitumor. Reprinted with permission from [65].

### 3.2. PDT-PTT for Synergistic Antitumor

In clinical practice, PDT typically produces unsatisfactory therapeutic results, which are hampered by the hypoxic microenvironment within solid tumors and the limited light penetration depth. Recently, PTT has been receiving more and more attention as an emerging and efficient mode of tumor treatment [66,67]. PTT uses light energy to kill cancer cells by converting it to heat energy via a nonradiative relaxation pathway. PTT does not face the drawbacks of being oxygen-dependent like PDT and has inherent advantages in the treatment of solid tumors in hypoxic environments. Therefore, the synergistic therapy of PTT and PDT could largely enhance the ablation capability of tumors in vivo. Wang et al. developed two NIR-II emission AIEgens (**CTBT** and **DCTBT**) via employing carbazole- or TPA-modified carbazole moieties as D, alkyl chain-modified thiophene groups as auxiliary D and π-bridge, and benzo [1,2-c:4,5-c’] bis ([1,2,5] thiadiazole) (BBT) as A, respectively (Figure 4a) [68]. The creation of strong D-A architecture facilitates narrowing the S_1_-S_0_ bandgap to achieve the long-wavelength absorption as well as emission, which also favors HOMO-LUMO separation. To achieve a small Δ*E*_ST_ and efficient ISC process, TPA was introduced as the rotors to nonradiatively dissipate the excited energy for heat generation. Moreover, the introduction of long alkyl chain on thiophene units contributes to providing the steric hindrance to twist the molecular geometry to improve the twisted intramolecular charge transfer (TICT) effect for **DCTBT**, thus further red-shifting the emission wavelength and accelerating the ISC processes. As a consequence, **DCTBT**-based nanoparticles showed predominate type I ROS generation and an excellent photothermal performance with a photothermal conversion efficiency (PCE) of 59.6%. After intravenous injection into tumor-bearing mice, **DCTBT** nanoparticles showed effective tumor accumulation at tumor sites. Benefitting from the synergistic cooperation of type I PDT and PTT, **DCTBT** exhibited excellent tumor inhibition performance on subcutaneous PANC-1 tumor-bearing mice as well as on the orthotopic pancreatic tumor-bearing mice.

The clever introduction of donor groups with strong electron-donating capability and large spatial spins can facilitate the ISC process as well as increase the nonradiative decay path of the aggregated state, thereby simultaneously improving the ROS generation capability and photothermal performance. Wang and coworkers reported four AIEgens for realizing a synergistic antitumor effect though type I PDT and PTT via an acceptor planarization and donor rotation strategy [69]. Thiophene-modified diketopyrrolopyrrole (**DPP**) was conjugated with the electron-donors via a metal-catalyzed cross-coupling reaction in previous works. This work utilizes methyl (as a new derivation site of **DPP**) to replace thiophene (traditional derivatization site of **DPP**), which was further modified through Knoevenagel condensation reaction to obtain **2TPAVDPP**, **TPATPEVDPP**, and **2TPEVDPP** (Figure 4b). The introduction of vinyl linkers as both sides of DPP could enlarge the acceptor planarity and π conjugation to facilitate the strong D-A interaction, thus promoting the ISC process as well as changing the type of PS pathways. As compared to the thiophene-linked **TPA-DPP**, which showed predominate type II ROS generation, the vinyl linked **2TPAVDPP**, **TPATPEVDPP**, and **2TPEVDPP** only showed type I ROS generation with negligible production of type II ROS. DFT calculation further revealed the T_1_ state energy level of **TPA-DPP** was located at 1.01 eV, while the T_1_ levels for the other three PSs were all below 0.77 eV. With the singlet oxygen energy level located at 0.98 eV, the lower T_1_ energy levels of **2TPAVDPP**, **TPATPEVDPP**, and **2TPEVDPP** made them unable to undergo energy transfer to ground state O_2_ to generate singlet oxygen, and hence, they mainly produced type I ROS. In contrast, the PCE values increased with the number of free rotating units, and **2TPEVDPP** nanoparticles possessed the highest PCE value of 66% among these analogues. As a consequence, **2TPEVDPP** nanoparticles achieved synergistic treatment of type I PDT as well as PTT under both normoxic and hypoxic environments. This molecular strategy of donor rotation and acceptor planarization provides a model for the development of AIE PSs with photothermal effects.

Multimodal imaging provides additional visualization for tumor treatment. As we all know, when photon energy is converted to heat, the resulting acoustic wave can be used for photoacoustic imaging (PAI) with increased penetration depth and signal-to-noise ratio. Moreover, PAI can be a powerful supplemental imaging approach to FLI, especially for NIR-II FLI, due to its advantages of clear contouring of deep tumor histology and clear microspatial resolution. Therefore, multimodal imaging of FLI and PAI will possess more potential for precision tumor treatment. Tang et al. recently developed three simple AIE-active phototheranostic agents (**TPEDCPy**, **TPEDCQu**, and **TPEDCAc**) with a D-A system and mitochondria-targeting ability through an electron acceptor engineering strategy for NIR II FLI/PAI guided diagnosis and efficient type I PDT and PTT combination phototherapy (Figure 4c) [70]. High twisted TPE and diphenylamine (DPA) moieties were constructed to the molecules as D and rotors. Furthermore, the electron-rich carbazole was employed as the π-bridge. With such a molecular design, the TICT effect was significantly increased, resulting in fluorescence emission from the NIR I region red-shifting to NIR II through enhancing the capacity of A moiety of the acceptors from quinolinium and pyridinium to acridinium. Moreover, this molecular design strategy can regulate the energy gap from 2.61 eV (**TPEDCPy**) to 2.33 eV (**TPEDCAc**), which makes **TPEDCAc** more conducive to accelerate the ISC process and enhance the type I ROS production capability. In addition, the large feature of acridinium improved the intramolecular motions; thus, **TPEDCAc** showed the highest PCE (44.1%) under the irradiation of 660 nm laser (0.3 W cm^−2^) among these analogues and other commercial photothermal agents (cyanine dyes ≈ 26.6% and ICG ≈ 3.1%) [71,72]. Importantly, **TPEDCAc** was successfully used in NIR II FLI/PAI guided PDT and PTT combination therapy on MCF 7 tumor bearing mice. Recently, Tang and coworkers reported three compounds (**TI**, **TSI**, and **TSSI**) for efficient multimodal imaging-guided tumor therapy (Figure 4d) [50]. The introduction of thiophene units as the π-bridge increased D-A interaction and promoted the ISC process and the type I ROS (OH^•−^) production. Additionally, **TSSI** also showed the best photothermal performance among these three analogues. Upon 660 nm laser irradiation (0.3 W cm^−2^, 5 min), the temperature of **TSSI** rapidly plateaued at 61 °C, higher than **TI** (47 °C) and **TSI** (54 °C). The PCE of **TSSI** nanoparticles was calculated to be ~46.0%, which provides a solid foundation for subsequent oncology treatment. The excellent photothermal conversion efficiency of **TSSI** nanoparticles also endows it with strong PA capability in vivo. Based on these advantages, **TSSI** nanoparticles are successfully used for multimodal imaging-guided PDT-PTT combination tumor therapy.

**Figure 4 molecules-28-00332-f004:**
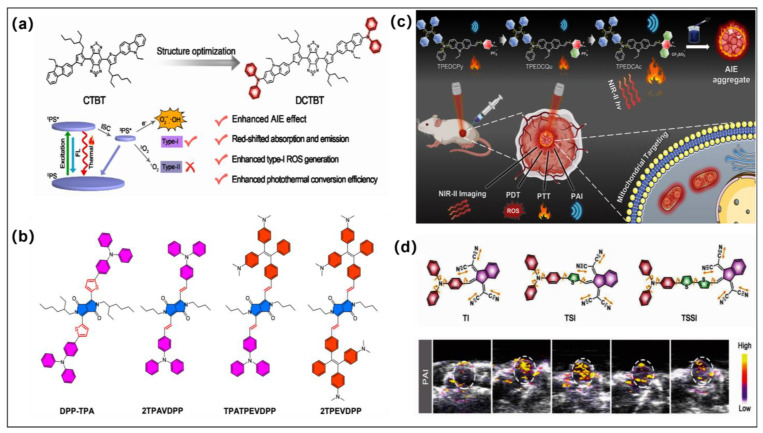
(**a**) Molecular structures and photoreaction mechanism of type I PSs (**CTBT** and **DCTBT**). Reprinted with permission from [68]. (**b**) Molecular structures of **2TPAVDPP**, **TPATPEVDPP**, and **2TPEVDPP**; fabrication of **2TPEVDPP** nanoparticles; and its NIR imaging-guided PDT-PTT. Reprinted with permission from [69]. (**c**) Molecular structures of **TPEDCPy**, **TPEDCQu**, and **TPEDCAc** and their NIR-imaging and PAI-guided PDT-PTT. Reprinted with permission from [70]. (**d**) Molecular structures of **TI**, **TSI**, and **TSSI** and MRI imaging-guided tumor therapy. Reprinted with permission from [50].

### 3.3. PDT-CDT for Synergistic Antitumor

Chemodynamic therapy (CDT) is similar to photodynamic therapy (PDT). It can generate ROS in the tumor microenvironment (TME) to kill tumor cells via external stimuli or endogenous triggers. The endogenous triggers are usually several kinds of transition metal ions, such as Fe, Cu, Mn, Co, etc., which are capable of transforming the endogenous H_2_O_2_ to the highly toxic OH^•−^ by Fenton or Fenton-like reactions under mildly acidic TME. More importantly, CDT, unlike PDT, does not require external stimuli and does not require the consumption of oxygen. Therefore, the synergistic treatment of PDT and CDT will effectively enhance the effect of tumor treatment with a lower dose and cost than PDT or CDT alone. Multifunctional nanoplatform development has been proposed as a promising strategy for effective PDT/CDT combination. In 2021, Wang and Tang et al. developed a smart TME-responsive multifunctional nanoplatform (MUM nanoparticles) for FLI-MRI guided PDT and CDT combination tumor therapy under both hypoxia-tolerance and deep-penetration conditions [73]. The powerful nanoplatform was constructed from type I AIE PSs (**MeOTTI**), MnO_2_ and upconversion nanoparticles (UCNPs) (Figure 5a). This nanoplatform realized triple-jump photodynamic theranostics: (1) Type I ROS generated by MUM nanoparticles under 980 nm laser irradiation. (2) The overexpressed GSH in the TEM can reduction MnO_2_ to Mn^2+^; subsequently, Mn^2+^ converts H_2_O_2_ to OH^•−^ through a Fenton-like reaction [74], and this process can be used as CDT. Furthermore, Mn^2+^ can also be used for T1-weighted MRI in cancer treatments [75]. (3) Type I ROS generated by **MeOTTI** under white light irradiation. MnO_2_ has the catalase-like capability, which can decompose H_2_O_2_ to O_2_, mitigating intracellular hypoxia (Figure 5b). Surprisingly, as shown in Figure 3c, the released Mn^2+^ can be used in cancer theranostics via T1-weighted magnetic resonance imaging (MRI). Furthermore, the excitation wavelength was shifted from the UV-vis region to the NIR region and was obtained using the FRET mechanism between **MeOTTI** and UCNPs, which obviously enhances the tissue penetration depth of phototherapy and OH^•−^ generation. This triple-jump PDT and CDT synergistic therapy strategy effectively inhibits tumor growth.

### 3.4. PDT-CDT-CT for Synergistic Antitumor

Chemotherapy, as one of the most common treatment strategies, is frequently associated with severe side effects and drug resistance, and patients experience excruciating pain, although it has a certain efficacy in tumor treatment. Therefore, combining CT with PDT-CDT will yield further therapeutic effects in oncology treatment. Recently, Wang and Tang developed a smart phototheranostic system via a multicomponent complementary-assembled strategy based on Cu^2+^-engineered aminosilica (Figure 6a,b) [76]. The tumor-targeted activatable aggregates (AD-Cu-DOX-HA) were prepared by coordinating previous reported AIE-active type I/II PS (**MeOTTVP**) [77] with Cu^2+^ (catalyze H_2_O_2_ to generate extremely poisonous OH^•−^ via the Fenton-like reaction) and further loading doxorubicin (DOX, as efficient drug for CT). Initially, the fluorescence of both **MeOTTVP** and DOX was in an “off” state because of the presence of Cu^2+^; however, after specific accumulation in tumor, hyaluronic acid (HA) in the surface layer of the aggregates was easily activated by acidic TME, further leading to stimuli-responsive PSs/DOX/Cu^2+^ release. The fluorescence signal of released PS (**MeOTTVP**) was recovered (over 10-fold) for accurate diagnosis and used in FLI-guided combinatorial therapy of type I PDT. The released Cu^2+^ further catalyzed H_2_O_2_ to generate highly toxic OH^•−^ for CDT. In addition, the released DOX was utilized for CT. The combination of these three treatment modalities has a significant effect on tumor growth inhibition. This work provides a new paradigm for smart and activable phototheranostic system.

### 3.5. PDT-Gas Therapy for Synergistic Antitumor

In recent years, gas (e.g., CO, NO, H_2_S) therapy has received increasing attention because these gases have few side effects and can be used as effective therapeutic agents [78,79,80,81,82]. These gases play a crucial role as endogenous signaling molecules in many physiological and pathophysiological events, and the combination of PDT with gas therapy is expected to further improve the efficacy of tumor treatment, especially when gas therapy is also initiated by light treatment. Very recently, Tang and Huang et al. developed a TSH hydrogel system for continuous type I ROS production after light irradiation for antitumor therapy application (Figure 6c,d) [83]. This multifunctional hydrogel platform was constructed by loading the (NH_4_)_2_S (a famed H_2_S donor) and type I AIE-active PS (**TDCAc**) into the injectable hydrogel. Moreover, **TDCAc** has a high PCE value of 43.5% and thermal stability, which plays a key role for its rapid heating under laser irradiation to soften TSH hydrogels for controllable release of (NH4)_2_S and **TDCAc** into the TME. Interestingly, the continual production of H_2_S by (NH4)_2_S in TME increases the amount of H_2_S that diffuses into cancer cells and inhibits the activity of catalase (CAT), effectively promoting the CDT effect (promote the Fenton-like reaction). In contrast, the uninterrupted H_2_O_2_ produced by **TDCAc** can bind the labile iron pool (LIP) in cells and promote the Fenton reaction to produce highly toxic hydroxyl radicals uninterrupted, which provides the demand of free radicals for subsequent tumor treatment. This work effectively enhances the tumor treatment effect and provides a new strategy of synergistic and efficient gas therapy based on type I AIE PSs.

**Figure 6 molecules-28-00332-f006:**
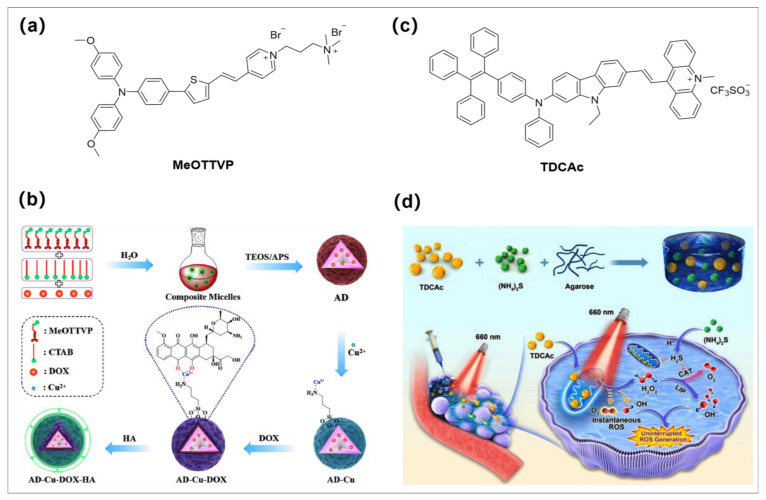
(**a**,**b**) are chemical structure of **MeOTTVP** and the schematic representation of the steps involved in making **AD-Cu-DOX-HA**, respectively. Reprinted with permission from [76]. (**c**) and (**d**) are the chemical structure of type I AIE PS (**TDCAc)** and schematic illustration of type I PDT combined with H_2_S gas treatment for indigenous LIP-mediated continuous OH^•−^ generation in the tumor, respectively. Reprinted with permission from [83].

### 3.6. PDT-Immunotherapy for Synergistic Antitumor

Immunotherapy has evolved into a promising cancer treatment strategy over the last few decades because it can assist the immune system in fighting cancer [84,85]. Immunogenic cell death (ICD) is a form of apoptotic cell death, providing an important theoretical rationale for modern clinical cancer immunotherapy [86]. Despite there being a limited number of PSs (e.g., pheophorbide A (PPa), Ce6, temoporfin, and hypericin) that can be employed as ICD initiators, these elicitors have not achieved satisfactory results for achieving ICD immunotherapy [87]. As a result, developing high-efficiency ICD initiators is critical for improving the efficacy of tumor immunotherapy. Ding’s group was the first to focus on mitochondrial oxidative stress and use AIE PSs to induce ICD in synergistic treatment of PDT and immunotherapy. Thereafter, numerous AIE PSs have been developed as ICD initiators to promote immunotherapy [88,89,90,91,92,93,94]. Although these reported AIE PSs (the majority of which are type II AIE PSs) are effective in initiating immunotherapy, the relationship between types of ROS and corresponding immune response is unknown. Very recently, several type I AIE-active PSs were successfully used in the synergistic treatment of tumors of PDT and immunotherapy.

Li et al. rationally developed three AIE type I PSs via the D-A effect for efficient facilitation of the reprogramming of macrophages to M1 phenotype for immunotherapy (Figure 7a) [45]. *n*-Butyl-substituted TPA and electron acceptors of different strengths (including ID, DCR, and BCI) were employed as the building blocks to construct the D-A structured molecules. **tTDCR** showed the smallest Δ*E*_ST_ with a value of 0.06 eV among these three analogues, which is far less than the appropriate value (<0.3 eV) for triggering the ISC process [58]. Hence, **tTDCR** displayed more efficient type I ROS generation capability than **tTDI** and **tTBCI**. The extracellular generation of ROS from **tTDCR** nanoparticles could significantly upregulate the secretion of typical proinflammatory cytokines (TNF-*α*, a famed marker of M1) from macrophages, with significantly higher activation than other experimental groups and control groups at all concentrations. Moreover, Western blot (WB) experimental results suggested the **tTDCR** nanoparticles possess the excellent capacity to downregulate CD206 (M2 marker) and upregulate phosphorylation of NF-*κ*B (M1 marker) (Figure 3c,d) over other analogues. All the experimental results clearly indicated that extracellular ROS generated by type I AIE PS can efficiently stimulate nonpolarized macrophages to M1 phenotype, and the stimulation efficiency improves with enhanced ROS generation ability (**tTBCI** < **tTID** < **tTDCR**). In addition, single treatment with **tTDCR** and light can achieve complete tumor ablation without recurrence within 20 days without the help of any immune adjuvants as the type I ROS overcomes the limitation of hypoxia and maintains highly efficient macrophage polarization even in anaerobic tumors (Figure 7e,f). Overall, this work indicates that highly effective type I AIE PSs provide new insights in PDT-mediated immunotherapy by inducing macrophage polarization.

Recently, Tang and coworkers developed a biomimetic nanoplatform (**CTTPA-G**) via loading a type I AIE PS (**TTPA**) and glutamine antagonist in cancer cell membranes (CC-Ms) as well as mesoporous silica nanoparticles (MSNs) for improving antitumor immunotherapy (Figure 7g,h) [95]. The D-A structured AIE PS was constructed by TPA moiety and dicycanovinyl-modified indanone moiety. O_2_^•−^and OH^•−^ are the main ROS species of **TTPA** upon light irradiation. Strong surface-exposed calreticulin (CRT) signaling and increased levels of extracellular high mobility group protein B1 (HMGB1) as well as adenosine triphosphate (ATP) secretion suggest that **CTTPA-G** can efficiently induce ICD process and activate DCs and specific T-cell responses. **CTTPA-G** stimulated the maturation of CD80^+^CD86^+^ of bone marrow dendritic cells DCs (BMDCs), indicating it can successfully activate the antitumor immune system. Flow cytometry results further indicated that type I ROS can effectively enhance the percentage of antitumor M1-like tumor-associated macrophages TAMs (CD11b^+^F4/80^+^CD86^+^), which also suggests that **CTTPA-G** can efficiently remodel the tumor immunosuppressive microenvironment. Ultimately, both the primary tumor and distal tumor growth were significantly inhibited (Figure 7i,j). In addition, tumor hypoxia was significantly relieved, mainly because **CTTPA-G** reduced cancer cell nutrition, improved TME, reshaped tumor metabolism, inhibited tumor proliferation, and obtained efficient antitumor immune responses. Moreover, **CTTPA-G** has a vaccine-like function that further synergistically inhibits tumor proliferation.

## 4. Type I AIE PSs for Antibacterial Applications

Bacteria are strongly tied to human existence and can affect human health in both positive and negative ways. [96]. Bacterial infections, particularly those that are resistant to antibiotics, increase morbidity and cause severe sickness, posing a substantial danger to global public health [97]. Although the emergence of antibiotics has given rise to hope for the management of bacterial infections, excessive or indiscriminate use of antibiotics often leads to the emergence of drug-resistant bacteria, making the disease more difficult to treat. As an emerging therapy without the involvement of antibiotics, PDT is gaining more and more attention because of its spatial and temporal selectivity, noninvasiveness, minimal resistance development, low side effects, and broad spectrum of antibacterial activity [98]. Moreover, the rapid development of AIE PSs has further promoted the use of PDT for bactericidal applications. Additionally, type I PDT is less oxygen-dependent than traditional PDT and has great potential for antibacterial therapy in a hypoxia environment since oxygen can be restored by intracellular SOD-mediated disproportionation. In this section, several different strategies are discussed for PDT antimicrobial applications of type I AIE PSs.

### 4.1. Single PDT Modality for Antibacterial

Efficient PSs, especially type I PSs for PDT sterilization, have been the greatest pursuit of scientific researchers. Given that bacterial surfaces are negatively charged, a wide range of AIEgens that are positively charged have been developed, which can further enhance the binding affinity to bacterial [99,100,101]. The cationic AIE PSs were developed for amplifying ROS generation, in particular type I ROS by Feng et al., and they were successfully used in PDT treatment of drug-resistant bacterial infection (Figure 8a) [55]. It is worth mentioning that the cationic PSs were favorable to enhance their binding affinity toward bacteria due to the nature of the negative charge of the bacteria cell membrane, which provides the basis for pathogen imaging and treatment of a wide range of bacteria and fungi. More interestingly, with the best type I ROS production ability and strong bacterial binding capability, **CTBZPyI** was successfully used in a methicillin-resistant staphylococcus aureus (*MRSA*)-infected skin wound model, and the type I ROS introduced by **CTBZPyI** upon light irradiation could eradicate *MRSA* and facilitate the healing of infected injuries with higher efficiency than the famed commercial antibiotics (vancomycin). This work offers a practical approach for developing the type I AIE PSs with efficient germicidal applications.

In the same year, Wang et al. reported a molecular engineering strategy by employing alkoxy-modified TPA and cationic pyridine moiety as D and A, respectively, to endow the PSs with an AIE feature, twisted molecular configuration, and charge separation characteristics (Figure 8b) [102]. The obtained **MTTTPy** possesses the highest type I ROS among these four analogues and an AIE emission feature, which makes it the most beneficial for photodynamic sterilization applications. Although it possesses poor germicidal efficacy toward Candida albicans (*C. albicans*) and *Escherichia coli* (*E. coli*), 95% of *S. aureus* can be eradicated by a low concentration of **MTTTPy** (0.5 μM) after exposure to white light (20 mWcm^−2^, 10 min). Gram-positive bacteria (*S. aureus*) animal model demonstrated that the fluorescence of **MTTTPy** can exist 96 h after injection, implying that **MTTTPy** can bind with bacteria for a long period of time without being quickly digested by blood circulation. More importantly, the photodynamic antibacterial effect was faster and more efficient than that of other control groups at 12 days after treatment. In a word, both in vitro and in vivo experimental results suggested that **MTTTPy** can be used in susceptible diagnostics and photodynamic treatment of Gram-positive bacterial.

Li et al. developed two planar AIEgens (**F-AB-DMA** and **DMA-AB-F**) as type I PSs for photodynamic treatment of multidrug-resistant bacteria (Figure 8c) [103]. According to the experimental data and theoretical calculation, the introduction of a fluorine substituent group could significant enhance AIE features, whereas substituent positions can alter the properties of ICT and the excited state double bond recombination (ESDBR). **DMA-AB-F** possessed stronger solid-state fluorescence emission and AIE characteristics since fluorine substituents enhance intramolecular hydrogen bonding and intensify intramolecular motion restriction. More interestingly, only **DMA-AB-F** has a proper and smaller energy system difference to facilitate the ISC process and, further, to generate type I ROS (OH^•−^) because of the distinct *E/Z*-configurational stacking behaviors. Importantly, **DMA-AB-F** was successful used in eliminating multidrug-resistant bacteria in vitro and in vivo due to its excellent type I PDT capability. This work breaks down the structural limitations of typical AIEgen designs and lays the foundation for the development of novel planar AIEgens for advanced biomedical applications.

### 4.2. PDT-CDT for Synergistic Antibacterial

To compensate for the shortcomings of PDT antibacterial, designing a combined PDT treatment strategy to effectively enhance antimicrobial efficacy is one of the strategies that researchers considered. Low pH (4.5–6.5) value and overexpressed H_2_O_2_ have been identified in the bacterial infectious microenvironment (IME) [104,105,106]. Therefore, with this feature of IME, the corresponding CDT can be developed for the elimination of bacteria. Gao et al. developed a composite (TPCI/MMT) by using iron-bearing montmorillonite (MMT) to deliver multicationic type I /II AIE PS (**TPCI**) to bacteria for an antibacterial effect (Figure 9) [107]. After TPCI/MMT and bacteria interact, the ^1^O_2_ and OH^•−^ produced by **TPCI** under light can be well used to perform PDT elimination of bacteria. In addition, the release of iron through MMT converts endogenous H_2_O_2_ into additional highly toxic OH^•−^ within the IME to simultaneously and continuously implement the CDT effect. The in vitro experimental results indicate that **TPCI**/MMT demonstrated a higher antibacterial efficacy under light irradiation than under darkness. It is worth mentioning that the bacterial survival rate of *E. coli* dropped to 43.7% after incubating **TPCI**/MMT with a very low concentration (0.5 mg·mL^−1^) in the dark. Moreover, in the light-exposed **TPCI**/MMT-treated *E. coli* solution, nearly all bacteria were eliminated, which suggests a significant PDT-CDT synergistic antibacterial effect. In vivo experimental results demonstrate that after treating with **TPCI**/MMT and light irradiation, the wound size reduced obviously, and the wound fully recovered, with the healing rate nearly 100% on day 14.

### 4.3. PDT-Gas Therapy for Synergistic Antibacterial

The healing of resistant keratitis remains an exceedingly difficult challenge because of the limited capability of current drugs to eradicate the continuous inflammatory phase of the cornea triggered by nuclear factor-κ-gene-binding (NF-*κ*B) as well as variable proinflammatory factors activated by bacterial endotoxin. Therefore, it remains extremely difficult to develop novel methods that have superior antimicrobial and anti-inflammatory properties to treat refractory keratitis. Since NO was shown to be widely used to eradicate broad-spectrum bacteria, including antibiotic-resistant bacteria, because it can downregulate NF-*κ*B involved in anti-inflammatory processes, more and more works focused on NO-based therapy [108,109,110,111,112]. Very recently, Ding et al. reported an NIR stimulus-responsive nanoplatform (UCNANs) for the on-demand release of NO and ROS for the synergistic therapies of refractory keratitis [113]. The nanoplatform (UCNANs) was constructed by the light-responsive core (UCNPs) and the shell (mesoporous silica) of loading type I AIE PS (**TPE-Ph-DCM**) and further grafted the bacterial-targeting moiety (COOH-PEG-QAC) and NO donor (**AMCNO-COOH**) (Figure 10a,b). Interestingly, when exposed to 808 nm light, UCNANs emits both UV and visible emission to induce NO release and activates AIE PS to produce type I ROS (O_2_^•−^). Moreover, NO can be used as a critical regulator of NF-*κ*B signaling inhibition and also can react with O_2_^•−^ to generate highly reactive ONOO^−^, further improving the antibacterial efficiency. More importantly, UCNANs possess anti-inflammation action because the eradication of bacteria decreased the virulence factors. Both in vitro and in vivo experimental results demonstrated that combined PDT and NO gas therapy could present a prospective strategy for the treatment of refractory keratitis.

## 5. Conclusions and Perspectives

Photodynamic therapy has become an emerging medical technique in addition to the traditional methods of treating cancer (i.e., surgery, immunotherapy, radiotherapy, or chemotherapy). In the aggregated state, the unique advantages of AIEgens such as enhanced fluorescence emission and ISC are particularly well suited for the development of efficient PSs for PDT. In this review, we first summarized effective strategies for promoting type I PDT by regulating ISC processes, mainly including D-A effect, polymerization, cationization engineering strategy, etc. These methods can further effectively improve the ROS production capacity of PSs, especially type I ROS, and can be used in biological applications such as directly or cooperatively killing tumors or bacteria. Subsequently, we also have discussed the recent developments in PDT and PDT-mediated synergistic treatment both in antitumor and antibacterial applications of type I AIE-active PSs.

Despite the fact that type I AIE-active PSs have advanced significantly in PDT in recent years, there are still many issues and difficulties that need to be overcome. Firstly, it is imperative to develop novel and more effective molecular design strategies to enhance the type I ROS generation capability. In addition, the excitation wavelength of most type I AIE PSs is in the ultraviolet or visible region, which seriously limits the treatment outcome of deep tissue disease; therefore, it is urgent to develop highly efficient NIR-absorbing type I AIE-active PSs. Sonodynamic therapy (SDT) can effectively enhance the efficacy of tumor treatment due to its noninvasive advantages of high tissue penetration depth (>10 cm). Thus, the synergistic treatment of PDT and SDT is promising to enhance the therapeutic efficacy of deep tissue disease. Traditional AIE PSs have almost “always on” fluorescence and ROS, making it difficult to ensure that normal tissues are not damaged by light after injection. However, if the PSs are designed to be activated by tumor or bacterial overexpressed substances (e.g., pH, GSH, H_2_O_2_, and related enzymes), the aforementioned problems can be effectively solved, and the accuracy and efficacy of diagnosis and treatment can be improved further; thus, it is necessary to accelerate the design of efficient activable type I AIE-active PSs. Last but not least, for unknown long-term biosafety, biodegradable type I AIE-active PSs are preferred, which can reduce biotoxicity and enhance metabolic capacity and further effectively promote clinical transformation in the future. We anticipate that this review will increase interest among researchers in the AIE field, especially in the creation of multifunctional type I AIE-active PSs for antitumor and antibacterial applications, and further facilitate clinical transformation.

## Figures and Tables

**Figure 2 molecules-28-00332-f002:**
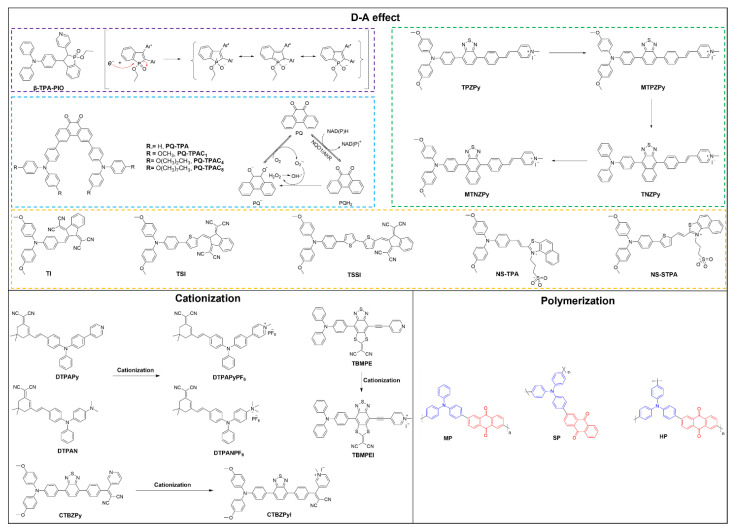
Strategies in enhancing type I ROS generation with exampled molecular structures.

**Figure 5 molecules-28-00332-f005:**
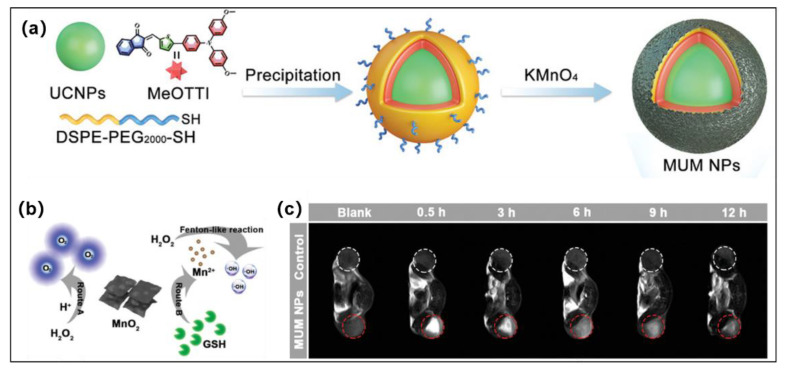
(**a**) Schematic illustration of chemical structures, nanofabrication of **MUM** nanoparticles. (**b**) Illustration of the effect of H_2_O_2_ and MnO_2_ in route A and route B. (**c**) Mice implanted with 4T1 tumors underwent in vivo T1-weighted MRI after being injected with MUM nanoparticle suspensions at various times. Reprinted with permission from [73].

**Figure 7 molecules-28-00332-f007:**
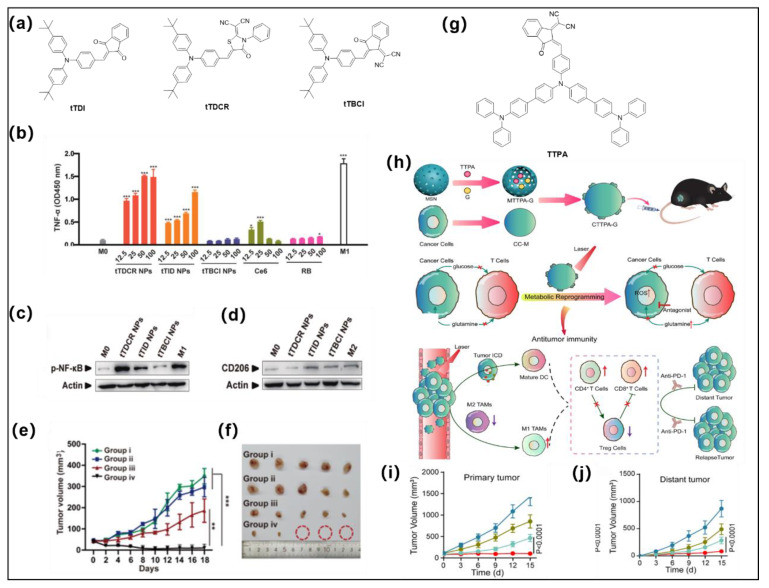
(**a**) Molecular structures of **tTDI**, **tTDCR**, and **tTBCI**. (**b**) The level of TNF-*α* secreted by M0 macrophage upon treatment with different PSs, * *p* < 0.05, *** *p* < 0.001. (**c**,**d**) are *p*-NF-κB and CD206 from cells treated with these AIE nanoparticles and white light irradiation, respectively. (**e**,**f**) Effect of inhibiting tumor growth under different treatment conditions. Reprinted with permission from [45]. (**g**) Molecular structures of **TTPA**. (**h**) Preparation route of **CTTPA-G** and its application in antitumor immune responses, tumor metastasis, and recurrence through tumor nutrient partitioning. (**i**,**j**) are the effects of inhibiting primary and distant tumor growth under different treatment conditions. Reprinted with permission from [95]. ** *p* < 0.01

**Figure 8 molecules-28-00332-f008:**
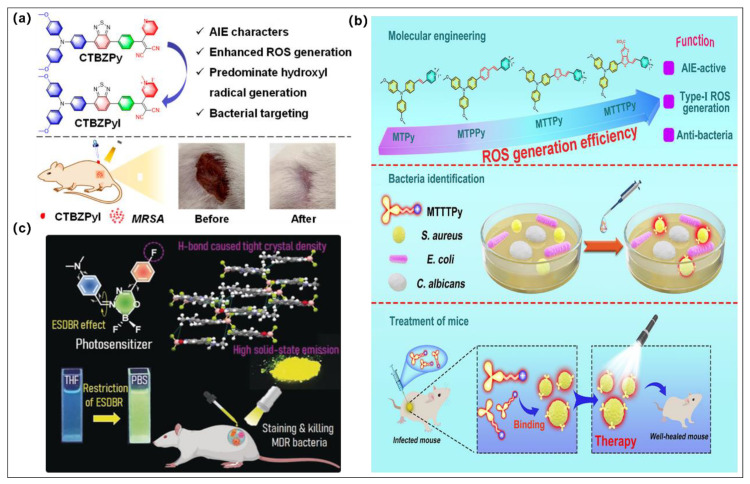
(**a**) Molecular structure of **CTBZPy** and **CTBZPyI** and the fast wound-healing effect of **CTBZPyI**. Reprinted with permission from [55]. (**b**) Molecular structures of four AIE PSs and schematic diagram of **MTTTPy**-mediated PDT for efficiently eradicating Gram-positive bacteria both in vitro and in vivo. Reprinted with permission from [102]. (**c**) Chemical structure of **DMA-AB-F** and schematic illustration of **DMA-AB-F** used for efficient staining and killing multidrug-resistant bacteria. Reprinted with permission from [103].

**Figure 9 molecules-28-00332-f009:**
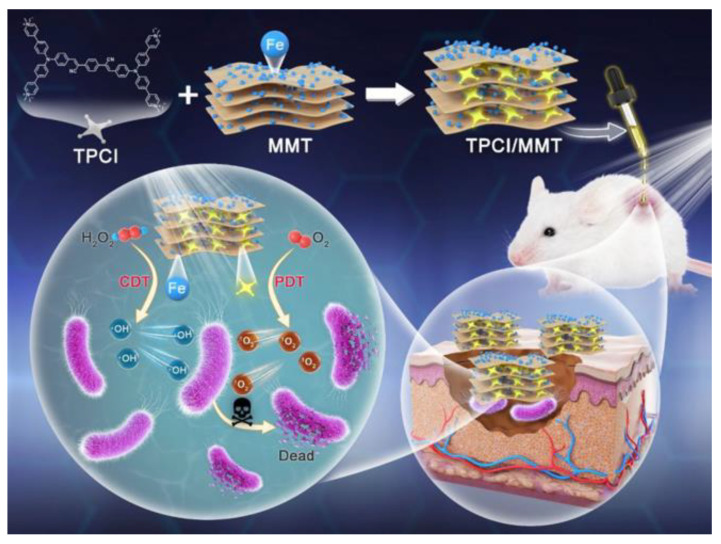
Molecular structure of **TPCI** and schematic illustration of the synergistic PDT-CDT based on **TPCI/MMT** for efficient eradication of bacterial. Reprinted with permission from [107].

**Figure 10 molecules-28-00332-f010:**
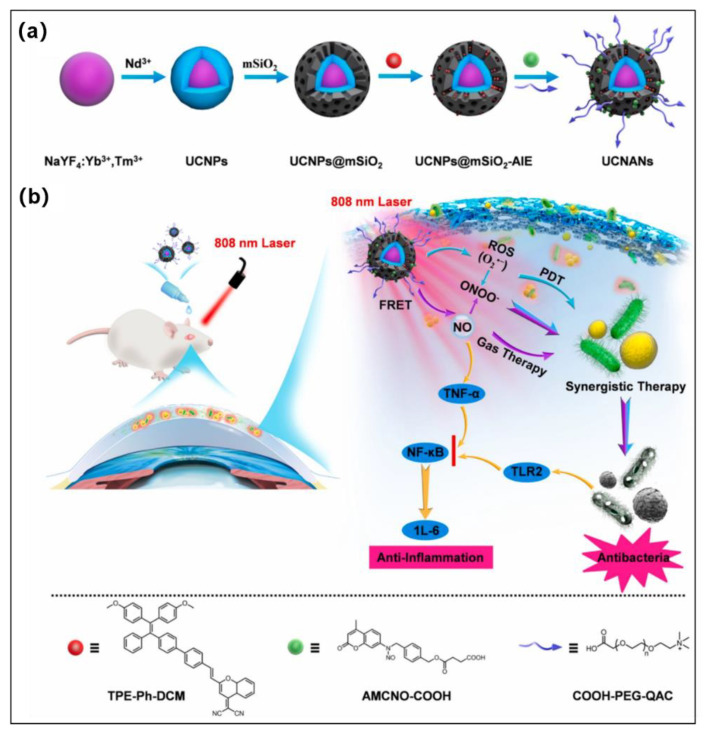
(**a**) Diagrammatic representation of the artificial pathway to **UCNANs**. (**b**) Illustration of **UCNANs** for synergistic type I PDT and NO gas therapy of refractory keratitis. Reprinted with permission from [113].

## Data Availability

Not available.

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
