# Peer review of "Recent Progress in Type I Aggregation-Induced Emission Photosensitizers for Photodynamic Therapy"

_molecules, 2022, doi:10.3390/molecules28010332_

Round 1
Reviewer 1 Report
The authors nicely summarized recent topics in type I photosensitizers for photodynamic therapy based on chromophores with aggregation-induced emission property. The contents are extensive and cover the entire area. This reviewer recommends publication of this manuscript in Molecules after addressing some errors in Figures:
1. p. 3: The resolution of Figure 2 appears very low. Please provide a higher-resolution figure. The larger size of chemical structures is preferred.
2. p. 7: In Figure 3a, the name of compound DTPAPyPF6 is covered. It would be better to show the chemical structures of DTPAPy and DTPAPyPF6 again in this panel.
3. p. 7: In Figure 3c, the labels of each panel are hard to read. They must be enlarged.
Author Response
Reviewer #1:
The authors nicely summarized recent topics in type I photosensitizers for photodynamic therapy based on chromophores with aggregation-induced emission property. The contents are extensive and cover the entire area. This reviewer recommends publication of this manuscript in Molecules after addressing some errors in Figures:
Reply: We sincerely thank the reviewer for recognizing our contribution and providing positive feedback to our work. We have addressed these issues raised by the reviewer.
- p. 3: The resolution of Figure 2 appears very low. Please provide a higher-resolution figure. The larger size of chemical structures is preferred.
Reply: According to the reviewer’s suggestion, we have modified the resolution of the picture and the size of the chemical structure.
- p. 7: In Figure 3a, the name of compound DTPAPyPF6 is covered. It would be better to show the chemical structures of DTPAPy and DTPAPyPF6 again in this panel.
Reply: We feel sincerely sorry for making such mistakes, and we have corrected these mistakes in the revised manuscript.
- p. 7: In Figure 3c, the labels of each panel are hard to read. They must be enlarged.
Reply: We feel sincerely sorry for making such mistakes, and according to the reviewer’s suggestion, we have enlarged the labels of each panel of the Figure 3c.

Reviewer 2 Report
This manuscript reviewed the latest advances in the reasonable design of AIE-active photosensitizers with type I photochemical mechanism for anticancer or antibacterial applications based on ISC modulation, as well as discuss the future prospects and challenges of them. This review is comprehensive and timely, which provides advice for the molecular design with the best energy conversion for improved PDT. The whole manuscript was logically organized and well written. I recommend the acceptance of this manuscript without changes.
Author Response
Reviewer #2:
This manuscript reviewed the latest advances in the reasonable design of AIE-active photosensitizers with type I photochemical mechanism for anticancer or antibacterial applications based on ISC modulation, as well as discuss the future prospects and challenges of them. This review is comprehensive and timely, which provides advice for the molecular design with the best energy conversion for improved PDT. The whole manuscript was logically organized and well written. I recommend the acceptance of this manuscript without changes.
Reply: We sincerely thank the reviewer for recongnition of our contribution and the positive feedback to our work.
